# A Meta-Analysis of the Effectiveness of Telemedicine in Glycemic Management among Patients with Type 2 Diabetes in Primary Care

**DOI:** 10.3390/ijerph19074173

**Published:** 2022-03-31

**Authors:** Anqi Zhang, Jinsong Wang, Xiaojuan Wan, Ziyi Zhang, Shuhan Zhao, Zihe Guo, Chufan Wang

**Affiliations:** 1School of Nursing, School of Public Health, Yangzhou University, Yangzhou 225000, China; mx120201109@stu.yzu.edu.cn (A.Z.); xjwan@yzu.edu.cn (X.W.); mz120201744@stu.yzu.edu.cn (Z.Z.); mx120201073@stu.yzu.edu.cn (S.Z.); mx120190874@stu.yzu.edu.cn (C.W.); 2Yangzhou Commission of Health, Yangzhou 225000, China; mx120201111@stu.yzu.edu.cn

**Keywords:** telemedicine, type 2 diabetes, primary health care, meta-analysis

## Abstract

**Introduction**: Telemedicine interventions are gradually being used in primary health care to help patients with type 2 diabetes receive ongoing medical guidance. The purpose of this study was to analyze the effectiveness of using telemedicine in primary health care for the management of patients with type 2 diabetes. **Methods**: A systematic search was conducted from database inception to August 2021 in nine databases, including PubMed, Web of Science, Cochrane Library, EMBASE, EBSCO, CNKI, Wanfang Data, VIP, and CBM. Data extraction and quality assessment were performed for studies that met the inclusion criteria. The meta-analysis was performed using Review Manager 5.4 (Cochrane) and Stata v.16.0SE (College Station, TX, USA). **Results**: A total of 32 articles were included in this study. Analysis showed a reduction in glycated hemoglobin, fasting glucose, and postprandial glucose after the telemedicine intervention. Systolic blood pressure and self-efficacy improved significantly, but there was no significant improvement in weight, lipid metabolism, or diabetes awareness. Subgroup analysis based on the duration of intervention showed significant improvement in glycated hemoglobin at 6 months of intervention. **Conclusions**: Telemedicine interventions may help patients with type 2 diabetes to effectively control blood glucose and improve self-management in primary health care. There is only moderate benefit, and the benefit may not be sustained beyond 6 months. However, the evidence for the improvement in lipid metabolism is insufficient and further studies are needed.

## 1. Introduction

The International Diabetes Federation 2021 reported that the number of people with diabetes mellitus (DM) has reached 533 million worldwide and is expected to reach 783 million worldwide by 2045. The prevalence rate is as high as 12.2% [1], of which type 2 diabetes (T2DM) [2] accounts for 90%. During the COVID-19 pandemic, the vulnerability of people with diabetes to public health emergencies became even more pronounced, with at least a two-fold increase in the risk of serious illness or mortality [3]. Telemedicine is a branch of e-health that uses communication networks to deliver healthcare services and health education to patients across regions [4]. Telemedicine is a viable alternative for patients seeking medical guidance for a chronic condition such as diabetes that requires repeated consultations with a physician, and without the risk of COVID-19 infection [5]. At the same time, telemedicine can provide patients with opportunities to connect and learn online and expand access to care through self-management and diabetes education [6]. It also allows for the involvement of both patients and the healthcare team to provide solutions to the patient’s needs [7]. Globally, the vast majority of people with T2DM are treated in primary health care settings [8]. Primary care is seen as the ideal point for making lifestyle changes [9]. Many countries have identified primary care telemedicine as a future development priority [10,11]. Therefore, this study aimed to analyze the effectiveness of telemedicine in primary health care for patients with T2DM.

Most previous studies on telemedicine for diabetes focused on the effect of telemedicine on the ability to self-manage diabetes [12], the impact of certain interventions on T2DM self-management [13] that are not specific to the primary health care setting [14], and other specific aspects [15]. Although there have been published reviews of the effectiveness of telemedicine research [16,17] on diabetes, there is a lack of analysis of all the potential outcomes and the impact of telemedicine interventions on patients with type 2 diabetes. In addition, telemedicine has grown rapidly in recent years and new research has emerged, thus requiring an updated review. Therefore, this study systematically reviewed the literature on telemedicine and evaluated the effectiveness of telemedicine technology for diabetes self-management with health outcomes among type 2 diabetes within the primary health care setting.

## 2. Methods

The meta-analysis was performed based on the recommendations of the Preferred Reporting Items for Systematic Reviews and Meta-Analyses (PRISMA) guidelines [18].

### 2.1. Search Strategy

The search was conducted in PubMed, Web of Science, Cochrane Library, EMBASE, EBSCO, CNKI, Wanfang Data, VIP, and CBM using a combination of subject terms, free terms, and MeSH terms from the time of database inception to 31 January 2021. Search terms included “type 2 diabetes”, “telemedicine”, “mobile health”, “primary health care”, and “community health services”. The review was registered prospectively (PROSPERO CRD42021299095). The list of search strategies is shown in detail in Appendix B. To prevent omissions, a snowball search was conducted for references of included articles and relevant reviews to supplement the relevant literature. No language restrictions were placed on the search terms.

### 2.2. Inclusion and Exclusion Criteria

The inclusion criteria for articles were as follows: (a) patients ≥18 years of age with T2DM; (b) since telemedicine is defined as the remote collection of records and transmission of patient data via telecommunications systems to healthcare providers for analysis and decision-making [19], email, smartphone, phone, SMS and text messages, web-based platforms, and hybrid forms (video calls and text messages) were included, without any face-to-face contact by a healthcare practitioner during the intervention; (c) improving one or more areas of diabetes self-management through telehealth; (d) control group using conventional treatment; (e) randomized controlled trial.

Exclusion criteria were (a) duplicate publications, (b) inaccessibility of full texts or extracted data, (c) studies without clinical data (abstract presentations, case reports, review articles, case-control, pilot studies, or study protocols), and (d) special populations such as children and pregnant women with T2DM.

### 2.3. Literature Screening and Data Extraction

Two researchers screened titles and abstracts that met the requirements based on the inclusion criteria, and when disagreements arose, a third researcher was consulted. If applicable, the disagreements were recorded to ensure a final consensus among the three reviewers. The literature searches were managed using EndNote X9 (Thomson ResearchSoft; Stanford, CT, USA). 

Two researchers independently extracted relevant data from the included studies using a self-designed form, including information on the author, year of publication, country, intervention method, and outcomes of both protocols in each of the clinical studies. In multiple-group studies, only telemedicine interventions and the conventional treatment groups were selected for comparison. For studies that could not be extracted or for which data were missing, attempts were made to contact the authors to obtain the data.

### 2.4. Methodological Quality Evaluation

The methodological quality of the included studies was independently evaluated by two investigators using the Critical Appraisal Skills Program (CASP) [20]. This list consists of 11 entries with three results: “yes”, “unclear”, and “no”. Disagreements, if any, were resolved through discussion with a third investigator or with the entire team.

### 2.5. Risk Bias Evaluation

Two researchers used Risk of Bias from the Cochrane Handbook for Systematic Review of Interventions 5.1.0 (2019) [21] to independently evaluate risk bias in the included studies. Bias included random sequence generation, allocation concealment, blinding of subjects and intervention providers, blinding of outcome evaluators, incomplete outcome data, selective reporting, and other sources of bias. Disagreements, if any, were resolved by discussion with a third author or by a discussion with the entire team.

### 2.6. Statistical Analysis

Meta-analysis was conducted using Review Manager 5.4 (Cochrane Collaboration, Oxford, UK) and Stata v.16.0SE (College Station, TX, USA) in this study. Continuous data were used with a mean and standard deviation as effect measures and estimates (MD) and 95% confidence intervals (CI) were given for each effect measure. We used the chi-square test (*p* < 0.01, statistically significant difference) and *I*^2^ to assess the heterogeneity between studies. When *I*^2^ < 50%, it indicated that there was no significant heterogeneity among studies, so a fixed-effect model was used; when *I*^2^ > 50%, it indicated that there was significant heterogeneity among studies, and a random-effects model was used. Sensitivity analysis was performed using a case-by-case exclusion approach to assess the stability of the study results. The presence of publication bias was assessed by a funnel plot and Egger and Begg tests when ≥10 studies were included with a two-sided significance level (*p* < 0.05, difference statistically significant). Subgroup analysis was used to observe the differences between intervention times.

## 3. Results

A total of 2460 articles were obtained through the search. After de-duplication, 1885 articles remained. After screening the titles and abstracts, a total of 147 studies were eligible. The PRISMA flow chart depicts the process of the literature screening (Figure 1).

### 3.1. Characteristics of Included Studies

Among the 32 included studies [22,23,24,25,26,27,28,29,30,31,32,33,34,35,36,37,38,39,40,41,42,43,44,45,46,47,48,49,50,51,52,53], 14 were from the United States, 3 from China, 2 from Spain, 2 from Australia, 2 from India, and 1 each from Australia, Iran, the United Kingdom, Slovenia, Sub-Saharan Africa, Canada, Belgium, Norway, Poland, and Italy. The studies ranged from 3 to 24 months of intervention between 2009 and 2021. Telemedicine in these studies used a variety of platforms to communicate and deliver the interventions, including cell phones (17.5%), Internet (17.5%), text messaging (27.5%), apps (20%), glucose-monitoring devices (12.5%), and tablets (5%). Some studies used a health belief model (13.3%), a trans-theoretical model of behavior change (26.7%), motivational interviewing (26.7%), social cognitive theory (13.3%), or chronic disease health management (20%) as part of the intervention. Intervention providers included physicians (18.9%), nurses (32.4%), allied health professionals (health education providers 5.5%; diabetes educators 2.7%; registered dietitians 2.7%; community workers 2.7%; telephone counselors 2.7%), and school research teams (32.4%). In addition, the content of routine care provided varied across studies, but in the majority of these studies, routine care provided general diabetes management. The details are shown in Table 1 and Table 2.

### 3.2. Literature Quality Evaluation and Risk Bias

Through the CASP inventory [20], the methodological quality of the included articles was assessed, and the results are presented in Table 3. Most of the studies were rated as moderate quality. Due to the specificity of the intervention, the vast majority of studies were not blinded to interventionists or the subjects.

All studies had clear reporting of sequence generation and were therefore considered to be low risk. Because there was no apparent allocation concealment, five [29,37,44,51,52] studies were rated as high risk. Twenty [23,24,25,27,28,29,31,33,34,35,36,37,44,45,46,47,50,51,52,53] studies were also rated as unclear bias because they were not blinded to interventionists or subjects. One [42] study was rated as high risk because it was not blinded to the outcome evaluator. Two [26,32] studies were rated as high bias risk because of incomplete outcome data. Figure 2 and Figure 3 show the results of the risk bias.

### 3.3. Results of the Meta-Analysis

#### 3.3.1. Primary Results (Effect on HbA1c)

Twenty-six studies [22,23,24,25,27,28,29,31,33,34,36,37,38,39,40,41,42,43,46,47,48,49,50,51,52,53] reported HbA1c outcomes at different intervention times. There were 2622 cases in the intervention group and 2666 cases in the control group. The results showed a statistically significant reduction in glycated hemoglobin levels in the intervention group compared to that in the control group (MD: −0.22, 95% CI [−0.34, −0.11], *p* < 0.0001; *I*^2^ = 51%, Figure 4). According to the time of intervention, subgroup analysis showed that in the mid-term intervention [22,25,27,28,36,37,39,41,47,49,52,53] (6–11 months) (MD: −0.36, 95% CI [−0.55, −0.16], *p* = 0.0003; *I*^2^ = 60%), glycated hemoglobin was reduced in the intervention group, but in the short-term intervention [33,42] (3–5 months), glycated hemoglobin did not change significantly (MD: −0.32, 95% CI [−0.68, 0.04], *p* = 0.08; *I*^2^ = 0%), and it may be diluted in the long-term intervention [23,24,29,31,34,38,40,43,46,48,50,51] (more than 12 months) (MD: −0.08, 95% CI [−0.21, 0.05], *p* = 0.21; *I*^2^ = 15%), as the results also did not change significantly. According to the means of intervention, subgroup analysis showed that in telemonitoring intervention [23,36,39,42,49,53] (MD: −0.33, 95% CI [−0.61, −0.05], *p* = 0.02; *I*^2^ = 58%, Figure 5) and application software [28,31,33,50] (MD: −0.26, 95% CI [−0.41, −0.11], *p* = 0.0007; *I*^2^ = 0%,), glycated hemoglobin was reduced in the intervention group, with statistically significant difference. However, in the three intervention methods of m-Health [24,25,29,39,46,51,52] (MD: −0.23, 95% CI [−0.50, 0.05], *p* = 0.10; *I*^2^ = 67%), telephone communication [27,34] (MD: −0.10, 95% CI [−0.34, 0.14], *p* = 0.40; *I*^2^ = 0%), and SMS [22,40,42,43,47,48] (MD: −0.09, 95% CI [−0.43, 0.25], *p* = 0.59; *I*^2^ = 65%), the results showed that these three intervention methods were superior to conventional nursing, but they are not significant. Both *p* = 0.637 obtained from Begg’s rank correlation test and *p* = 0.213 from Egger’s linear regression method suggested no significant publication bias. The funnel plot is shown in Appendix A. 

#### 3.3.2. Secondary Results

##### Impact on Fasting Blood Glucose

Eight studies [36,37,42,43,44,46,47,52] reported the results of fasting blood glucose (FBG). There were 538 cases in the intervention group and 545 cases in the control group, and the duration of intervention ranged from 3 to 24 months. Meta-analysis showed a statistically significant reduction in FBG levels in the intervention group compared to that in the control group (MD: −0.49, 95% CI [−0.86, −0.12], *p* = 0.01; *I*^2^ = 36%, Figure 6).

##### Effects on Postprandial Blood Glucose

Four studies [37,41,44,52] reported the results of post-prandial blood glucose (PBG). There were 297 cases in the intervention group and 298 in the control group, and the duration of the intervention ranged from 3 to 6 months. Meta-analysis showed a statistically significant difference in reduced PBG levels in the intervention group compared to that in the control group (MD: −2.06, 95% CI [−2.80, −1.31], *p* < 0.00001; *I*^2^ = 36%, Figure 7).

##### Effect on Body Weight

Nineteen studies [22,24,28,29,31,34,35,37,38,40,43,44,46,47,48,49,50,52,53] reported the results of body weight. The duration of intervention ranged from 3 to 24 months in 2418 cases in the intervention group and 2461 cases in the control group. Meta-analysis showed no significant difference in body weight between the intervention and control groups (MD: −0.12, 95% CI [−0.36, 0.12], *p* = 0.33; *I*^2^ = 16%, Figure 8). Begg’s rank correlation test yielded *p* = 0.319 and Egger’s linear regression method yielded *p* = 0.506, both suggesting no significant publication bias. The funnel plot is shown in Appendix A.

##### Effect on Systolic Blood Pressure

Thirteen studies [25,28,29,34,43,44,47,48,49,50,53] reported the results of systolic blood pressure. There were 1847 cases in the intervention group and 1853 cases in the control group, and the duration of the intervention ranged from 3 to 24 months. Meta-analysis showed that systolic blood pressure improved in the intervention group compared to that in the control group, with a statistically significant difference (MD: −2.43, 95% CI [−3.44, −1.42], *p* < 0.00001; *I*^2^ = 0%, Figure 9). Begg’s rank correlation test yielded *p* = 0.155 and Egger’s linear regression method yielded *p* = 0.191, both suggesting no significant publication bias. The funnel plot is shown in Appendix A.

##### Effects on High-Density Lipoprotein, Low-Density Lipoprotein, Triglycerides, and Total Cholesterol

Ten studies [24,28,42,43,46,47,49,50,52,53] reported high-density lipoprotein (HDL) results, 11 studies [24,28,29,34,42,43,46,47,50,53] reported low-density lipoprotein (LDL) results, 6 studies [28,40,47,50,52] reported triglyceride (TG) results, and 10 studies [28,29,34,38,42,43,46,47,50,52,53] reported total cholesterol (TC) results. All four results were not significantly different (Figure 10). For HDL, Begg’s rank correlation test yielded *p* = 0.371 and Egger’s linear regression method yielded *p* = 0.238. For LDL, Begg’s rank correlation test yielded *p* = 1.000 and Egger’s linear regression method yielded *p* = 0.266. The above suggested no significant publication bias. The funnel plots are shown in Appendix A.

##### Effects on Self-Efficacy, Diabetes Knowledge, Physical Health, and Mental Health

There were three separate outcomes reporting self-efficacy, diabetes knowledge, physical health, and mental health, in which only self-efficacy improved in the intervention group compared to the control group (MD: 0.34, 95% CI [0.16, 0.52], *p* = 0.0002; *I*^2^ = 0%; Figure 11).

##### Effects on Exercise, Foot Care, and Satisfaction

There were three separate outcomes reporting exercise, foot care, and satisfaction, in which only exercise improved in the intervention group compared to the control group (MD: 0.75, 95% CI [0.27, 1.22], *p* = 0.002; *I*^2^ = 48%; Figure 12). 

## 4. Discussion

This study confirms the effectiveness of various telemedicine interventions applied in primary health care. Meta-analysis showed that despite high heterogeneity, telemedicine interventions can improve patients’ metabolic control more than conventional controls over 6 months, as is supported in other meta-analyses [54,55,56]. However, we did not find any significant effect on lipid metabolism such as body weight, HDL, TG, or TC, which is different from the results of several previous studies [55,57,58]. This may be because, on the one hand, most of the study interventions focused on the improvement of patients’ blood glucose, and fewer studies provided a full range of interventions on patients’ diet, exercise, and diabetes knowledge. One the other hand, lipid control in diabetic patients is mainly dependent on the use of drugs [59], so there is no difference between the two.

### 4.1. Blood Sugar Control

The results of the study showed that the telemedicine intervention resulted in a significant decrease in clinical glycated hemoglobin levels, with a mean decrease of 0.1% to 0.32%. HbA1c concentrations effectively reflected the patients’ average blood glucose levels over the previous 8–12 weeks. Thus, the reduction in HbA1c concentrations reflects the positive impact of telemedicine on the long-term care of patients with T2DM. In the subgroup analysis regarding duration, the intervention group showed lower HbA1c than the conventional group only when the intervention duration was 6 months. In contrast, there was no significant improvement in glycosylated hemoglobin at short-term (3 months) or long-term (12 months) intervention times. This may be because T2DM is a slow-treating chronic disease and the HbA1c level reflects the average blood glucose over 2–3 months [60]. Therefore, there was no significant improvement in glycated hemoglobin for the short-term intervention, which is consistent with other studies [61,62]. In addition, the small number of studies of short-term interventions included in the articles of this study may be somewhat biased, and hence, more and higher quality studies may be needed for further evaluation. The decline in benefit at 12 months of intervention may be related to the patient’s response to the intervention, which is consistent with the results of other studies [63,64]. Therefore, in studies of long-term interventions, patient engagement declines with the duration of the intervention, and continuous monitoring and individualized reminders are important to maintain the continuity of the intervention.

### 4.2. Self-Management

Good self-management can significantly reduce the occurrence and progression of T2DM complications [65]. In this study, we evaluated the patients’ self-management abilities based on results of FBG and PBG, and the results showed that the blood glucose level was reduced in the intervention group compared to the control group, as is supported in other meta-analyses [66]. Moreover, significant improvements in self-efficacy occurred in the intervention group. Ten of these studies [22,29,30,33,35,38,39,40,42,47], combined with theoretical models, improved the self-management ability of patients with T2DM by providing them with knowledge about blood glucose monitoring, diet, and exercise, and by providing them with timely personalized message reminders through the data obtained. However, since the number of each study is limited, it is not possible to determine whether there are interactions between different telemedicine technologies (e.g., text messaging, smartphones). Most research trials reported on glycosylated hemoglobin concentrations, and only a small number of study trials reported on other aspects regarding diabetes management (FBG, postprandial glucose, lipid metabolism, self-management ability, etc.). Therefore, great care should be taken in interpreting the results of other secondary outcomes. The meta-analysis also showed that telemedicine strategies were associated with other clinical benefits of glycemic control and that telemedicine measures contributed to improved self-management abilities, including systolic blood pressure and diabetes self-efficacy, in patients with T2DM.

### 4.3. Telemedicine

In the study by Natalie Robson et al. [67], phone-based telehealth interventions had the greatest impact on the self-management of people with type 2 diabetes. In the study of Puikwan A. Lee et al. [68], although the results of meta-analysis showed that telemedicine intervention was better than usual care, it was impossible to judge which intervention method would benefit patients the most. The results of this study showed that telemedicine intervention was superior to conventional care. In the subgroup analysis of different intervention methods, only remote monitoring and application software intervention significantly improved HbA1c in the intervention group, whereas there were no significant differences in mobile medicine, telephone communication, or SMS remote intervention. Interventions delivered through remote monitoring had the greatest impact on type 2 diabetes-related outcomes, followed by app interventions, M-Health, telephone counseling, and SMS interventions. At the same time, there was significant heterogeneity among individual studies, which may have been caused by the different methods and applications of telemedicine intervention. Therefore, it is difficult to determine which interventions are most effective in managing type 2 diabetes. A total of five [22,29,33,39,48] studies assessed patient engagement or perception of telemedicine in this review. Although these studies reported widespread patient acceptance of telemedicine interventions, only three studies formally tested subjects’ satisfaction. Meta-analysis results showed that although the satisfaction of the intervention group was better than that of the control group, the difference was not statistically significant. At the same time, high-frequency and short-term interventions [25,41,47,53] can significantly improve the hemoglobin of patients. However, the effectiveness of the intervention decreases with the duration of the intervention, so the next step that should be considered is how to maintain a high level of patient compliance during prolonged intervention. Only one study [48] reported the costs associated with the intervention and no cost-benefit analysis was performed, making it impossible to draw firm conclusions about the cost-effectiveness of telemedicine interventions in type 2 diabetes.

In response to the ongoing COVID-19 pandemic, telemedicine has seen incremental growth [69], with one study [44] reporting improved patient glycemic control during the COVID-19 pandemic with telemedicine-based patient interventions using an app. We believe this number is too small to draw firm conclusions about telemedicine in diabetes care in the context of COVID-19. Although telemedicine can provide benefits to patients, many problems remain in its future development. For example, the professional and ethical challenges posed by telemedicine may affect both patients and physicians [70]. Patients’ trust in telemedicine interventions may be reduced due to untimely phone calls or incorrect text messages. Moreover, patient compliance decreased with the prolongation of intervention time. In addition, in most countries the cost of telehealth interventions is borne by the patient, which can lead to low patient acceptance. Therefore, future research should explore a convenient and timely telemedicine system, improve the quality of intervention, and increase patient acceptance through the training of intervention teams, as well as maintain a certain frequency of intervention to increase patient compliance, ensure government funding to reduce telemedicine costs, and accelerate telemedicine adoption.

## 5. Limitations and Strengths

First, the study found high heterogeneity in some outcomes, and the high heterogeneity may have been due to differences in telemedicine interventions, subjects, the timing of interventions, interventionists, and intervention frequency. Second, because we found a small number of studies reporting diabetes self-management ability, the findings are inconclusive regarding the effectiveness of self-management and require large sample testing and further investigation. Third, there are fewer studies with long-term follow-up, and the findings may contain only short-term effects of improving T2DM and lack evaluation of long-term effects (development of complications, mortality, etc.). In addition, the cost-effectiveness of the intervention becomes uncertain as the duration of the intervention increases. Fourth, the nature of the intervention led to the fact that most of the studies were not blinded to the interventionists or subjects. Fifth, the inclusion of studies from different countries resulted in significant differences in baseline primary health care systems. For example, government-funded telemedicine services are common in primary health care in the United States and Europe [71], but in some developing countries, especially South Africa, even basic services are not available due to low government budgets [72]. This may affect the compliance of patients. Sixth, because the term “telemedicine” is very nonspecific, conclusions may not be generalizable. Finally, we identified slight missing corners in some of the funnel plots, which may have been caused by the presence of small sample sizes and low-quality studies in our included studies. However, the Egger and Begg tests revealed that there was no significant publication bias in the included studies, and sensitivity analyses did not show any significant change in the effect values of the study results. Therefore, this does not affect our conclusions.

This meta-analysis has multiple strengths. First, we assessed the quality of evidence for all included articles using the CASP [20] method. Second, we did not restrict the search language, so articles in all countries and languages were eligible for this study. Third, we strictly followed the recommendations of the meta-analysis evaluation method [73,74,75]. Fourth, we had no time limit on previous published articles, so no earlier published articles were missed.

The meta-analysis of this study supports that telemedicine interventions can be effective in improving glycemic control in patients with T2DM in primary health care. This information is critical for decision-makers to develop interventions based on desired outcomes and to maximize the use of resources in the primary health care setting.

## 6. Conclusions

The results of this study suggest that compared with non-telemedicine care, telemedicine intervention in primary health care may improve glycemic control and self-management in patients with type 2 diabetes. In particular, the six-month intervention period was most effective, and interventions with remote monitoring or apps were most effective. Therefore, telemedicine interventions are an effective supplement to traditional face-to-face counseling in primary health care settings, and telemedicine can provide convenience for homebound patients with T2DM who do not have timely access to medical resources. The next step should focus on assessing the acceptability and feasibility of telemedicine implementation on a large scale and whether it is effective in reducing healthcare costs.

## Figures and Tables

**Figure 1 ijerph-19-04173-f001:**
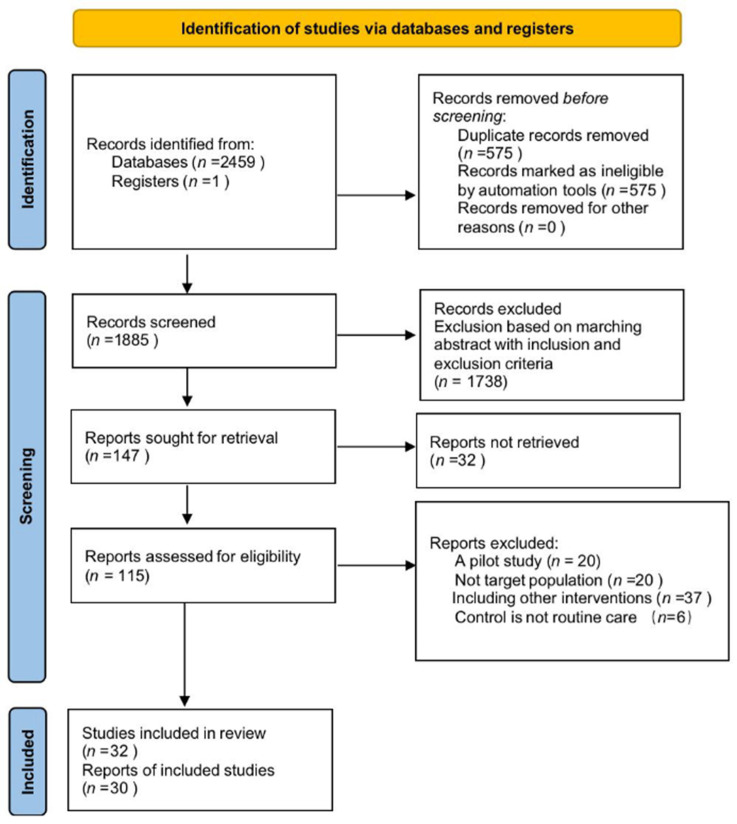
Flow diagram of preferred reporting items for systematic reviews and meta-analyses.

**Figure 2 ijerph-19-04173-f002:**
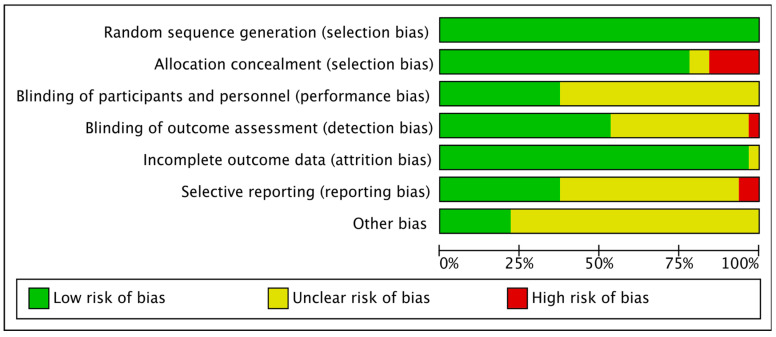
Overall summary of risk of bias in the included studies.

**Figure 3 ijerph-19-04173-f003:**
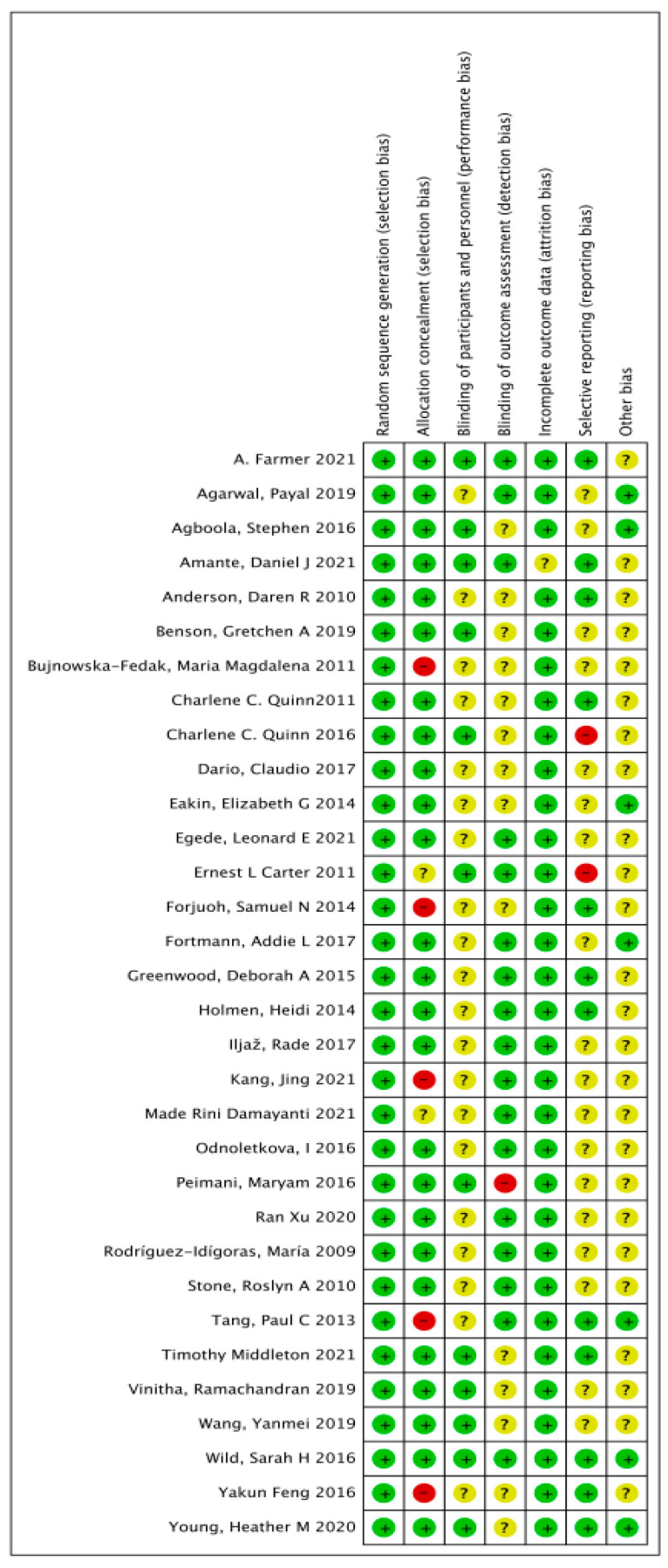
Risk of bias in the included studies. **+:** Low risk of bias; -: High risk of bias; ?: Unclear risk of bias.

**Figure 4 ijerph-19-04173-f004:**
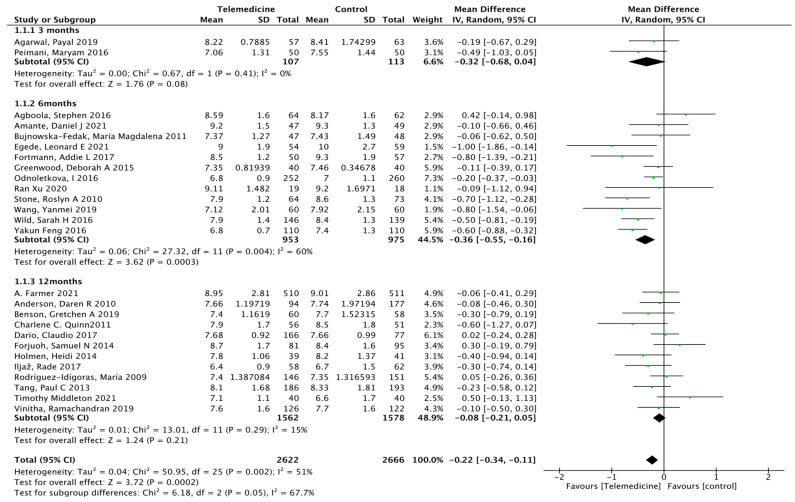
Subgroup analysis showed higher HbA1c reduction at 6 months of intervention.

**Figure 5 ijerph-19-04173-f005:**
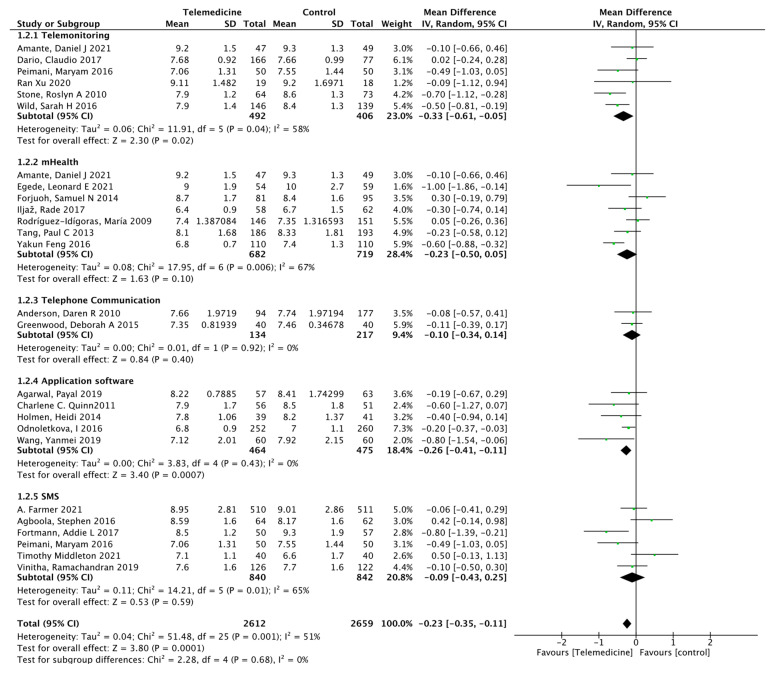
Subgroup analysis showed higher HbA1c reductions in patients with remote monitoring and application interventions.

**Figure 6 ijerph-19-04173-f006:**
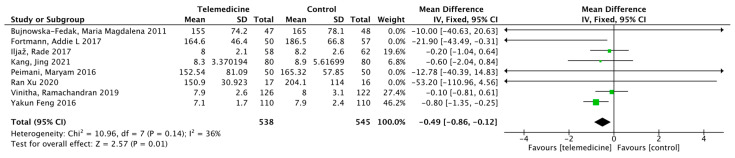
Mean difference in the changes in FBG levels for telemedicine and usual care interventions.

**Figure 7 ijerph-19-04173-f007:**
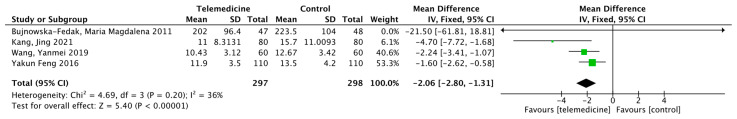
Mean difference in the changes in PBG levels for telemedicine and usual care interventions.

**Figure 8 ijerph-19-04173-f008:**
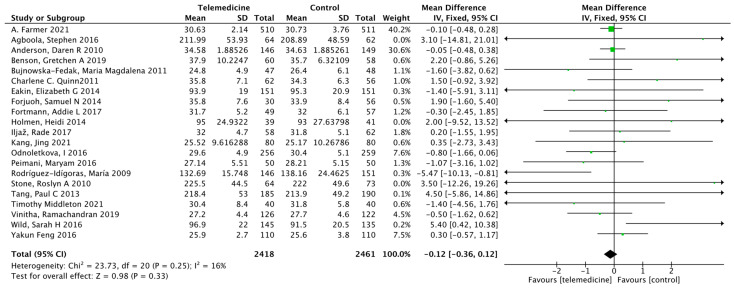
Mean difference in the changes in weight levels for telemedicine and usual care interventions.

**Figure 9 ijerph-19-04173-f009:**
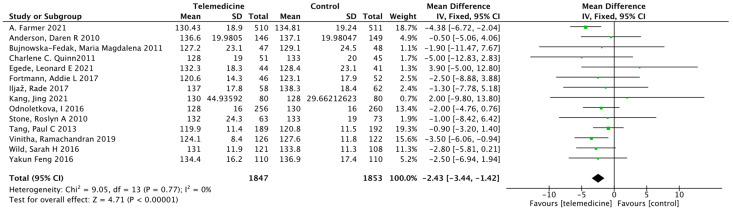
Mean difference in the changes in SBP levels for telemedicine and usual care interventions.

**Figure 10 ijerph-19-04173-f010:**
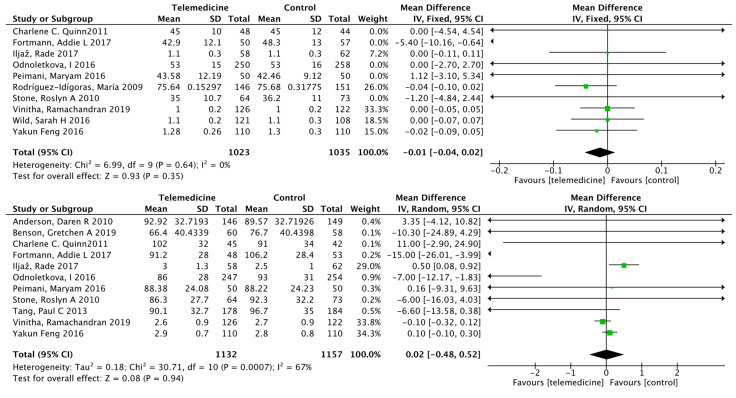
Mean difference in the changes in HDL, LDL, TG, and TC levels for telemedicine and usual care interventions.

**Figure 11 ijerph-19-04173-f011:**
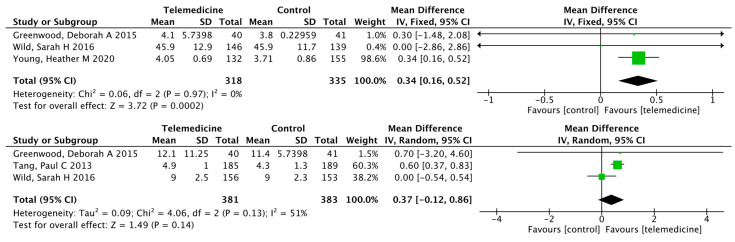
Mean difference in the changes in self-efficacy, diabetes knowledge, physical health, and mental health levels for telemedicine and usual care interventions.

**Figure 12 ijerph-19-04173-f012:**
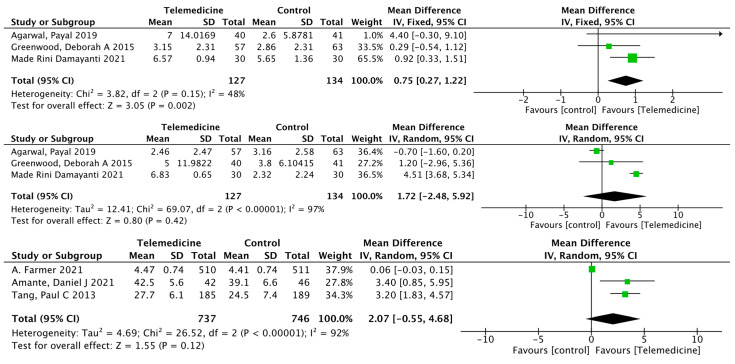
Mean difference in the changes in exercise, foot care, and satisfaction levels for telemedicine and usual care interventions.

**Table 1 ijerph-19-04173-t001:** Characteristics of telemedicine studies.

Author (Year) Country	Interveners	Intervention Basis	Intervention Method	Routine Group	Intervention Time; Frequency	Primary Outcome	Secondary Outcome	Intervention Method						
								Education	Feedback	Counseling	Goals	Prompt	Motivation	Monitoring
Stephen Agboola (2016) USA [22]		Health literacy concepts, the transtheoretical model of behavior change	SMS, pedometers (monitoring and uploading data)	Reminder telephone calls for those participants who did not upload their activity data after 5 consecutive days, and usual care.	6 months; 2 text messages daily	Mean step counts (collected by the wireless pedometers)	HbA1c, weight, physical activity stage of change questionnaire, usability and satisfaction	Y	Y			Y	Y	Y
Daniel J. Amante (2021) USA [39]		AADE National Standards for Diabetes Self-Management Education curriculum	SMBG, the Livongo Care Team of CDEs would contact participants by their preferred communication method (either phone call or text message) within 3 min of receiving an abnormal SMBG notification from the Smart Cloud.	Usual care	6 months; within 3 min after receiving the abnormal SMBG notification	Changes in HbA1c during each time period	Diabetes Treatment Satisfaction Questionnaire (DTSQ)					Y		Y
Charlene C. Quinn (2011) USA [50]	Doctor		Mobile diabetes management software application and a web portal	Usual care	12 months; real time based on feedback	HbA1c	Patient Health Questionnaire-9 (PHQ), Diabetes Distress Scale, Self-Completion Patient Outcome Instrument, diabetes complications (blood pressure, lipid levels)	Y	Y					Y
Daren R. Anderson (2010) Spain [34]	EHR nurses		Telephonic disease management	Usual care	12 months; patients were called weekly, bi-weekly, or monthly depending on their risk stratification.	HbA1c	BMI, SBP, DBP, and LDL	Y						Y
Rade Iljaž (2017) Slovenia [46]	Health-care providers		e-Diabetes application: upload data and send automatic alerts via simple email and text messages	Usual care	12 months; data were recorded every two weeks. These reminders were sent if a user had not entered body weight, blood pressure, physical activity, and diet data within 2 weeks of the deadline or not completed the COOP-WONCA questionnaire within 8 weeks.	Change from baseline of HbA1c at 1 year	HbA1c at 6 months, BMI, COOP-WONCA Questionnaire, blood lipids, SBP, DBP	Y		Y		Y		
Ramachandran Vinitha (2019) India [43]			Health education SMS	Usual care	24 months; 2–3 educatory text messages per week	HbA1c	FPG, 2hPG, lipid parameters, weight, waist circumference, blood pressure, physical activity, quality of life and dietary aspects, acceptability of text messages	Y						
Samuel N. Forjuoh (2014) USA [51]			a. CDSMP(chronic disease self-management program); b. PDA(diabetes self-care software); c. a combination of both interventions (CDSMP + PDA)	Usual care	12 months; enter every day	HbA1c	BMI and Blood pressure, along with several self-management behavioral measures (e.g., foot care)							Y
Deborah A. Greenwood (2015) USA [27]	Diabetes educators		In-home tablet computer: telehealth remote monitoring system	Usual care	6 months; every day	HbA1c	Diabetes Knowledge Test (DKT), Summary of Diabetes Self-Care Activities (SDSCA), Diabetes Empowerment Scale short form (DES-SF)	Y				Y		Y
Jing Kang (2021) China [44]	Doctor, nurse		WeChat app	Usual care	3 months; twice a week	FBG, PBG, BMI, blood glucose, TIR, and blood pressure		Y		Y		Y		Y
Maryam Peimani (2016) Iran [42]		Social cognitive theory	a. Tailored SMS group; b. non-tailored SMS group	Usual care	3 months; 7 messages per week	HgA1C levels, FBS, lipid profile, BMI, Self-Care Inventory (SCI), Diabetes Management Self-Efficacy Scale (DMSES)		Y	Y				Y	
Paul C. Tang (2013) USA [29]	Nurse-led, multidisciplinary health team	Universal models of behavior change, motivational interviewing techniques, Chronic Care Model	The PAMF Online-mediated Personalized Health Care Program, which couples a multidisciplinary diabetes care management team with an EHR-integrated Online Disease Management (ODM) system	Usual care	12 months; periodically uploading data	HbA1c	Blood pressure, LDL, 10-year Framingham cardiovascular risk, satisfactionand psychosocial well-being	Y	Y					Y
Payal Agarwal (2019) Canada [33]		Transtheoretical Model of Behavior Change	BlueStar mobile app: customized, evidence-based messages are delivered in real time based on information uploaded by patients.	Usual care	3 months; every day	HbA1c	Patient-reported outcomes measures (PROMs), patient-reported experience measures (PREMs), EuroQol-5D (EQ-5D), Problem Areas in Diabetes (PAID), The Summary of Diabtes Self-Care Activities Measure (SDSCA-6), availability	Y					Y	Y
Gretchen A. Benson (2019) USA [38]	Registered dietitian nutritionist (RDN)	Health Belief Model, Transtheoretical Model	Phone coaching intervention: medical nutrition therapy (MNT), combined with pharmacotherapy	No intervention	12 months; intervention calls generally lasted about 30 min, with a specific call frequency tailored to patient preferences.	The composite number of diabetes optimal care goals met	BMI, LDL, Morisky scale	Y	Y		Y			
Maria Magdalena Bujnowska-Fedak (2011) Poland [37]	Nurse		Telehome monitoring system	Usual care	6 months; upload data at least once a week and receive a text message if you exceed the alarm line.	Regular glucometry, HbA1c, blood cell count, erythrocyte sedimentation rate, cholesterol balance, body mass index, creatinine concentration, urine analysis, blood serum electrolytes, blood pressure	Quality of life, doctor–patient communication, sense of control over the disease					Y		Y
Claudio Dario (2017) Italy [23]			Telehealth service: Patients were equipped at home with a glucometer and a gateway for data transmission to a Regional eHealth Center (ReHC).	Usual care	12 months; every day	HRQoL: SF-20 questionnaire	HbA1c, outpatient, emergency, hospitalization rates, bed days of hospital care, Hospital Anxiety and Depression Scale (HADS)							Y
Addie L. Fortmann (2017) USA [47]		Culturally appropriate DSME curriculum	Dulce Digital participants received up to motivational, educational, and/or call-to-action SMS.	Usual care	6 months; two to three messages a day were sent at study start, with frequency tapering over 6 months.	HbA1c and lipids (TC, LDL, HDL, and TG), SBP, DBP, body weight and height, self-report items, feasibility, acceptability		Y				Y		Y
Heidi Holmen, MSc (2014) Norway [31]	GPs, health providers		a. Few Touch Application (FTA): diabetes diary app; b. the FTA with health counseling (FTA-HC): increase health counseling	Usual care	12 months	HbA1c	Health Education Impact Questionnaire (heiQ), Short-Form 36v2 Health Survey (SF-36), Center for Epidemiologic Studies Depression Scale (CES-D)	Y	Y			Y		Y
Yanmei Wang (2019) China [41]	Nurse		Mobile application: health monitoring, health guide, gealth advice, follow-up	Usual care	6 months; patients upload data on a daily basis and the nurses evaluate them 2 to 3 days a week.	Glycemic control compliance rate, self-management ability of patients with diabetes, questionnaire on disease awareness, rehospitalization rate and number of hospital visits		Y	Y	Y		Y		Y
Sarah H. Wild (2016) UK [49]	Nurse		Telemonitoring and glycemic control	Usual care	9 months; one fasting and one nonfasting blood glucose at least twice weekly and BP and weight measured at least weekly	HbA1c	Ambulatory systolic, DBP, weight, anxiety, depression, quality of life, self-efficacy, self-reported physical activity, self-reported exercise tolerance, self-reported alcohol intake, diabetes knowledge		Y					Y
I Odnoletkova (2016) Belgium [28]	Nurse		COACH Program: an update of the best practice guidelines for the management of type 2 diabetes, motivational interviewing techniques and software program use	Usual care	6 months; an average of 5 times over 6 months, each time 10–45 min	HbA1c	TC, LDL, HDL, TG, blood pressure, BMI, self-perceived health status, diabetes-specific emotional distress, satisfaction, annual healthcare utilization	Y	Y				Y	Y
Heather M. Young (2020) USA [30]	Nurse	Motivational interviewing	Wearable tracking device (Basis Peak, then Garmin VivoSmart Heart Rate), telephone	Usual care	9 months; conference call every 2 weeks for 3 months, real-time data upload	Diabetes self-efficacy (Diabetes Empowerment Scale [DES]–Short Form)	Depression severity (Patient Health Questionnaire-9 (PHQ-9), physical function, emotional distress, anxiety, Perceived Stress Scale (PSS)	Y			Y		Y	Y
Elizabeth G. Eakin (2014) Australia [35]	Telephone counselor	Motivational interviewing, social cognitive theory	Telephone counseling	Usual care	18 months; 4 initial weekly calls, fortnightly calls for 5 months, monthly calls for 12 months	Weight, accelerometer-derived MVPA, HbA1c	Dietary energy intake, diet quality, waist circumference, fasting blood lipid levels, blood pressure	Y		Y	Y		Y	
Roslyn A. Stone (2010) USA [53]	Nurse		ACM + HT: home telemonitoring coupled with active medication management by a nurse practitioner	Monthly telephone calls from the study’s diabetes nurse educator	6 months; patients upload data daily and nurses provide timely phone contact.	HbA1c, blood pressure, weight, a fasting lipid panel	Medication regimen (dose), changes in the regimen (dose and date)	Y		Y				Y
Yakun Feng (2016) China [52]	Doctor, nurse		U-Healthcare: Patients upload data and doctors give feedback over the phone or on the web.	Usual care	6 months; 2 times a week	HbA1c, FPG, 2 hPG, TG, TC, HDL, LDL, BMI, SBP, DBP, BUN, Scr, AST, ALT, r-GT		Y	Y					Y
Leonard E. Egede (2021) USA [25]	Nurse case manager, doctor		Web-based TACM intervention: the FORA 2-in-1 Telehealth System for diabetes to link a case manager to poorly controlled diabetics in real time.	Usual care	6 months; patients upload data on a daily basis and nurses adjust patients’ medication weekly or biweekly under the supervision of doctors.	BP, QOL: 12-item Short-Form Health Survey (SF-12)								Y
María I. Rodríguez-Idígoras (2009) Spain [24]	Doctor, nurse		Telemedicine system: possibility of sending the SMBG values of the patients to a web page via mobile SMS messages. The HCP had a password access to this web page to check the blood glucose values of the patients and if necessary send to them SMS messages with recommendations.	No intervention	12 months; send text messages if necessary	HbA1c	Blood glucose, TC, HDL, LDL, TG, BMI, SBP, DBP, and system adherence			Y		Y		Y
A. Farmer (2021) Sub-Saharan Africa [48]		Capability, opportunity, Motivation Behavior Model	SMS	Usual care	12 months; three to four times a week	HbA1c	Systolic blood pressure, Lipids, EuroQol 5-Dimension 3-Level (EQ-5D-3L), cardiovascular risk and the proportion of participants reaching treatment goals					Y	Y	Y
Timothy Middleton (2021) Australia [40]			SMS: personalized support and reminder programs based on text messages	Usual care	12 months; two a week for the first two months; one per week in the third month; one per month after the fourth month	All scheduled follow-up appointments	Overall clinic attendance, HbA1c, BMI, total cholesterol, triglycerides, diabetes self-management practices	Y				Y	Y	
Made Rini Damayanti (2021) India [45]			SMS	Usual care	10 weeks; three times a day	Diabetes Self-Care Activities Measure		Y				Y	Y	
Ran Xu (2020) USA [36]			Self-reported FBG data were collected by EpxDiabetes automated phone calls or text messages.	Usual care	6 months; three times a week.	HbA1c, FBG	Response rate, engagement rate					Y		Y
Ernest L. Carter (2011) USA [26]	Nurse		Laptop, wireless scale, blood pressure cuff, glucometer: Upload biometric data, make action plans with nurses through video conferencing, and watch related health videos.	Usual care	9 months; every two weeks.	HbA1c, BMI, blood pressure, qualitative interview		Y		Y				
Charlene C. Quinn (2016) USA [32]	Doctor		Software application, glucose meter and glucose testing supplies: Upload blood glucose monitoring data, receive health education information, and check health files.	Usual care	12 months; real time based on feedback	HbA1c		Y				Y		Y

HbA1c: glycated hemoglobin A; FBG: fast blood sugar; 2hPG: 2hr plasma glucose; BMI: body mass index; SBP: systolic pressure; DBP: diastolic pressure; HDL: high-density lipoprotein cholesterol; LDL: low density lipoprotein cholesterol; TC: total cholesterol; TG: triglycerides; BUN: blood urea nitrogen; AST: aspartate aminotransferases; ALT: alanine transaminase; Scr: serum creatinine; r-GT: gamma-glutamyltransferase; app: application; SMS: short message service; SMBG: self-monitoring of blood glucose.

**Table 2 ijerph-19-04173-t002:** Characteristics of telemedicine intervention.

Author (Year)	Diabetes Management	Diet	Exercise	Diabetes Complications (Eye, Foot)	Diabetes Symptoms (Hypoglycemia)	Medication Compliance	Other	Diabetes Managemen	Diet	Exercise	Diabetes Complications (Eye, Foot)	Diabetes Symptoms (Hypoglycemia)	Medication Compliance	Other
	Intervention Group							Control Group						
Stephen Agboola (2016) [22]	Y	Y	Y			Y		Y						
Daniel J. Amante (2021) [39]	Y	Y	Y					Y						
Charlene C. Quinn (2011) [50]	Y	Y	Y	Y	Y	Y		Y						
Deborah A. Greenwood (2015) [27]	Y	Y	Y			Y		Y					Y	
Jing Kang (2021) [44]	Y	Y	Y	Y	Y		Psychological counselling	Y	Y	Y				
Maryam Peimani (2016) [42]	Y	Y	Y			Y	Blood glucose monitoring	Y						
Paul C. Tang (2013) [29]	Y	Y	Y		Y	Y		Y						
Yanmei Wang (2019) [41]	Y	Y	Y	Y	Y	Y		Y	Y	Y			Y	
Sarah H. Wild (2016) [49]	Y					Y	Lifestyle change	Y						
Heather M. Young (2020) [30]	Y	Y	Y				Self-efficacy	Y						
Daren R. Anderson (2010) [34]	Y	Y	Y			Y		Y						
Rade Iljaž (2017) [46]	Y	Y	Y					Y						
RamachandranVinitha (2019) [43]	Y	Y	Y	Y		Y		Y	Y					
Payal Agarwal (2019) [33]	Y	Y	Y	Y	Y			Y						
Gretchen A. Benson (2019) [38]	Y	Y				Y								
Maria Magdalena Bujnowska-Fedak (2011) [37]	Y	Y				Y		Y						
Claudio Dario (2017) [23]							Blood glucose monitoring							
Addie L. Fortmann (2017) [47]	Y				Y	Y		Y						
Yakun Feng (2016) [52]	Y	Y	Y	Y	Y		Lifestyle and weight control	Y						
Elizabeth G. Eakin (2014) [35]	Y	Y	Y				Weight loss	Y						
Leonard E. Egede (2021) [25]	Y					Y		Y						
Roslyn A. Stone (2010) [53]	Y					Y		Y						
María I. Rodríguez-Idígoras (2009) [24]	Y						Blood glucose monitoring	Y						
Heidi Holmen, MSc (2014) [31]	Y	Y	Y					Y						
I Odnoletkova (2016) [28]	Y	Y	Y	Y	Y	Y		Y			Y			
Samuel N. Forjuoh (2014) [51]	Y	Y	Y			Y		Y						
Timothy Middleton (2021) [40]	Y	Y				Y		Y	Y					
A. Farmer (2021) [48]	Y					Y		Y						
Made Rini Damayanti (2021) [45]	Y	Y	Y			Y		Y						
Ran Xu (2020) [36]	Y						Blood glucose monitoring	Y						
Ernest L. Carter (2011) [26]	Y	Y	Y	Y	Y			Y						
Charlene C. Quinn (2016) [32]	Y	Y	Y	Y	Y	Y		Y						

**Table 3 ijerph-19-04173-t003:** Critical Appraisal Skills Program (CASP) checklist on the included studies.

Study	Q1	Q2	Q3	Q4	Q5	Q6	Q7	Q8	Q9	Q10	Q11
Stephen Agboola et al. (2016) [22]	Y	Y	Y	Y	Y	L	Y	Y	?	Y	?
Daniel J. Amante et al. (2021) [39]	Y	Y	Y	Y	Y	L	Y	Y	N	Y	Y
Charlene C. Quinn et al. (2011) [50]	Y	Y	Y	?	Y	S	Y	Y	N	Y	Y
Daren R. Anderson et al. (2010) [34]	Y	Y	Y	N	Y	S	Y	Y	N	Y	Y
Rade Iljaž et al. (2017) [46]	Y	Y	Y	?	Y	S	Y	Y	N	Y	Y
RamachandranVinitha et al. (2019) [43]	Y	Y	?	Y	Y	L	Y	Y	N	Y	Y
Samuel N. Forjuoh et al. (2014) [51]	Y	?	Y	N	Y	S	Y	Y	N	Y	?
Deborah A. Greenwood et al. (2015) [27]	Y	Y	Y	?	Y	S	Y	Y	?	Y	Y
Jing Kang et al. (2021) [44]	Y	N	?	?	Y	L	Y	Y	?	Y	Y
Maryam Peimani et al. (2016) [42]	Y	Y	Y	N	N	L	Y	Y	?	Y	Y
Paul C. Tang et al. (2013) [29]	Y	N	Y	?	Y	S	Y	N	?	Y	Y
Payal Agarwal et al. (2019) [33]	Y	Y	Y	?	Y	S	Y	Y	Y	Y	?
Gretchen A. Benson RDN et al. (2019) [38]	Y	Y	Y	?	Y	N	Y	Y	?	Y	Y
Maria Magdalena Bujnowska-Fedak et al. (2011) [37]	Y	N	?	N	Y	S	Y	N	?	Y	Y
Claudio Dario et al. (2017) [23]	Y	Y	Y	N	Y	S	Y	Y	?	Y	Y
Addie L. Fortmann et al. (2017) [47]	Y	Y	?	?	Y	S	Y	N	?	Y	Y
Heidi Holmen, MSc et al. (2014) [31]	Y	Y	Y	?	Y	S	Y	Y	N	Y	?
Yanmei Wang et al. (2019) [41]	Y	Y	Y	?	Y	S	Y	N	?	Y	Y
Sarah H. Wild et al. (2016) [49]	Y	Y	Y	Y	Y	S	Y	N	N	Y	Y
I Odnoletkova et al. (2016) [28]	Y	Y	Y	?	Y	S	Y	Y	Y	Y	Y
Heather M. Young et al. (2020) [30]	Y	Y	Y	?	Y	S	Y	N	?	Y	Y
Elizabeth G. Eakin et al. (2014) [35]	Y	Y	?	N	Y	S	Y	Y	N	Y	?
Roslyn A. Stone et al. (2010) [53]	Y	Y	Y	?	Y	L	Y	Y	N	Y	Y
Yakun Feng et al. (2016) [52]	Y	?	Y	N	?	S	Y	N	N	Y	Y
Leonard E. Egede et al. (2021) [25]	Y	Y	Y	?	Y	S	Y	Y	?	Y	Y
María I. Rodríguez-Idígoras et al. (2009) [24]	Y	Y	Y	?	Y	N	Y	Y	N	Y	Y
A. Farmer et al. (2021) [48]	Y	Y	Y	Y	Y	L	Y	Y	Y	Y	Y
Timothy Middleton et al. (2021) [40]	Y	Y	Y	?	Y	L	Y	Y	?	Y	Y
Made Rini Damayanti et al. (2021) [45]	Y	Y	Y	N	Y	S	N	Y	Y	Y	Y
Ran Xu et al. (2020) [36]	Y	Y	Y	N	Y	S	Y	Y	?	Y	Y
Ernest L. Carter et al. (2011) [26]	Y	Y	Y	?	Y	L	Y	Y	N	Y	Y
Charlene C. Quinn et al. (2016) [32]	Y	Y	?	N	Y	S	Y	N	?	Y	Y

?: Cannot tell; L: large; N: no; S: small; Y: yes.

## Data Availability

Not applicable.

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
