# Peer review of "A Meta-Analysis of the Effectiveness of Telemedicine in Glycemic Management among Patients with Type 2 Diabetes in Primary Care"

_ijerph, 2022, doi:10.3390/ijerph19074173_

Round 1

Reviewer 1 Report

The authors have carried out a meta-analysis of the effectiveness of telemedicine in glycemic management among patients with type 2 diabetes in primary care.  The methodology described is well described.

Please mention if a protocol for doing the metanalysis was agreed upon. I note it was registered with PROSPERO. The analysis performed is appropriate. The comparator for most studies is usual care and so did the authors notice any difference based on the frequency of contact in the active intervention.

Given that most of the studies were of moderate quality with a high degree of heterogeneity,  unblinded intervention the conclusion that "telemedicine intervention can ( may is a better word) significantly (too strong) improve glycemic control and self-management ability but not lipid metabolism in patients with T2DM over 6 months. Lipid control in diabetes is mainly driven by the use of medication, I am not surprised there is no difference with telemedicine.

It looks like there could be a moderate benefit and the benefit may not be sustained beyond 6 months.  Given that only certain interventions were effective they should be highlighted both in the abstract and the conclusion.

A relevant study that shows the benefits, but also highlights the limitations and the caveats. However it is important to recognise the word Telemedicine is very nonspecific and the conclusions are not generalisable

Author Response

Thank you very much for the valuable comments from the reviewers. The corresponding responses are as follows:

Point 1: Please mention if a protocol for doing the mete-analysis was agreed upon. I note it was registered with PROSPERO. The analysis performed is appropriate. The comparator for most studies is usual care and so did the authors notice any difference based on the frequency of contact in the active intervention. 

Response 1: As you mentioned, there is some variability in intervention frequency across all studies, however, analysis based on intervention frequency may not be possible due to differences in interventions, which may account for the high heterogeneity and is one of the limitations of this paper one. Therefore, based on your suggestion, we have revised and highlighted the limitations of line 368 of the article. Thank you for your suggestion.

Point 2: Given that most of the studies were of moderate quality with a high degree of heterogeneity,  unblinded intervention the conclusion that "telemedicine intervention can ( may is a better word) significantly (too strong) improve glycemic control and self-management ability but not lipid metabolism in patients with T2DM over 6 months. Lipid control in diabetes is mainly driven by the use of medication, I am not surprised there is no difference with telemedicine.

Response 2: As suggested, In lines 272-281, we have revised and highlighted the words in the discussion. Thank you for your suggestion.

Point 3:It looks like there could be a moderate benefit and the benefit may not be sustained beyond 6 months. Given that only certain interventions were effective they should be highlighted both in the abstract and the conclusion.

Response 3: As suggested, We have added and highlighted lines 24-25 and 401-402. for the duration of the intervention and the intervention regimen in the conclusion of the summary and conclusion .Thank you for your suggestion.

Point 4: A relevant study that shows the benefits, but also highlights the limitations and the caveats. However it is important to recognise the word Telemedicine is very nonspecific and the conclusions are not generalisable.

Response 4:  As suggested, We have revised and highlighted the limitations in lines 381-382. Thank you for your suggestion.

Reviewer 2 Report

Thank you for the opportunity to review this high-quality work, assessing the impact of telemedicine on glycemic control and other parameters of medical follow-up of people with type 2 diabetes. The analysis is methodologically sound and the manuscript very well-written and informative.

I have only some minor comments that could be addressed before acceptance.

  1. Introduction: The introduction is very well written and informative and adequately places the context of the research. However, the authors refer only to the benefits / positive aspects of telemedicine in diabetes. Are there any negative aspects or risks that physicians and patients should take into account (e.g., inability for physical examination which could lead to wrong diagnosis)?
  2. It would be very interesting to see a sub-analysis that included studies performed during the COVID-19 pandemic, since the latter largely affected and changed the way we use telemedicine in the follow-up of people with diabetes.

Author Response

Thank you very much for the valuable comments from the reviewers. The corresponding responses are as follows:

Point 1: Introduction: The introduction is very well written and informative and adequately places the context of the research. However, the authors refer only to the benefits / positive aspects of telemedicine in diabetes. Are there any negative aspects or risks that physicians and patients should take into account (e.g., inability for physical examination which could lead to wrong diagnosis)?

Response 1: As suggested, we have added and highlighted the impact of telemedicine on patients and physicians in the context of lines 355-356. Thank you for your suggestion.

Point 2: It would be very interesting to see a sub-analysis that included studies performed during the COVID-19 pandemic, since the latter largely affected and changed the way we use telemedicine in the follow-up of people with diabetes.

Response 2: Based on your comments, We re-categorized all included articles and found that only one of the interventions studied was an intervention study conducted during the COVID-19 epidemic, and an intervention using a similar application to the previous study, for COVID-19 Less evidence was provided on the effectiveness of telemedicine interventions during the period. And due to the small number of eligible intervention studies, classification based on this criterion may not be possible. As suggested, We have revised as suggested as highlighted on lines 350-353 line. Thank you for your suggestion.

Reviewer 3 Report

Results of this systematic revise (SR) and meta-analysis support the potential of telemedicine in the treatment of persons with type 2 diabetes .Overall, methods and results are clear and appropriate, and the Discussion is thorough. I have several suggestions and questions for clarification.

Lines 51-52: By “lack of analysis of the overall outcomes” do you mean that not all potential outcomes have been evaluated in other meta-analyses? This needs a little more clarity.  

Line 55: I suggest changing “determined” to “evaluated”. Also, include the outcomes in the statement of purpose, e.g., “…. evaluated the effectiveness of telemedicine technology for ______ in the primary health setting.”

Several columns in Table 1 could be combined to make a little more room: Author, Year, and Country could be in one column; Intervention time (suggest using duration) and frequency could be combined. The orientation of the table printout was portrait – perhaps the layout could be landscape to make more space for each column. It’s awkward to read as it is now. The same applies to Table 2.

Please label all Figures as to what is being presented. One should be able to interpret the results presented in the figure without having to go to the text.

 Line 136: By “cross-theoretical model of change”, do you mean “trans-theoretical model of change”? If so, suggest changing.

Line 265 and Line 368: I suggest changing “confirms” to “supports”.

Lines 294-298: Did any other SRs and meta-analyses look at these outcomes? If so, were results consistent or different?

Line 319: It would be helpful to include which outcomes differed in the previous studies. All of them or several of them?

Line 347:  What are the strengths of the study?  Suggest including these and changing heading to “Strengths and Limitations”.

Line 359: By differences in primary health care systems, do you mean what comprises conventional care, what the individual has to pay, other? Please clarify.

Line 361-362: It is unclear what is meant by “acceptance rate”. Of telemedicine? Is it participation or compliance to treatment? Both?

Line 376:  Just to clarify, in the included studies, was all treatment (including initial treatment) provided via telemedicine in the intervention groups of each study? From the inclusion criteria in the methods, this is my interpretation. If conventional care was provided in-person/face to face with telemedicine follow-up, were those studies excluded?  I am just wondering if the conclusion is that telemedicine a good alternative, or is it also a potentially cost-effective means to provide follow-up vs in-person counseling?

Results of this systematic revise (SR) and meta-analysis support the potential of telemedicine in the treatment of persons with type 2 diabetes .Overall, methods and results are clear and appropriate, and the Discussion is thorough. I have several suggestions and questions for clarification.

Lines 51-52: By “lack of analysis of the overall outcomes” do you mean that not all potential outcomes have been evaluated in other meta-analyses? This needs a little more clarity.  

Line 55: I suggest changing “determined” to “evaluated”. Also, include the outcomes in the statement of purpose, e.g., “…. evaluated the effectiveness of telemedicine technology for ______ in the primary health setting.”

Several columns in Table 1 could be combined to make a little more room: Author, Year, and Country could be in one column; Intervention time (suggest using duration) and frequency could be combined. The orientation of the table printout was portrait – perhaps the layout could be landscape to make more space for each column. It’s awkward to read as it is now. The same applies to Table 2.

Please label all Figures as to what is being presented. One should be able to interpret the results presented in the figure without having to go to the text.

 Line 136: By “cross-theoretical model of change”, do you mean “trans-theoretical model of change”? If so, suggest changing.

Line 265 and Line 368: I suggest changing “confirms” to “supports”.

Lines 294-298: Did any other SRs and meta-analyses look at these outcomes? If so, were results consistent or different?

Line 319: It would be helpful to include which outcomes differed in the previous studies. All of them or several of them?

Line 347:  What are the strengths of the study?  Suggest including these and changing heading to “Strengths and Limitations”.

Line 359: By differences in primary health care systems, do you mean what comprises conventional care, what the individual has to pay, other? Please clarify.

Line 361-362: It is unclear what is meant by “acceptance rate”. Of telemedicine? Is it participation or compliance to treatment? Both?

Line 376:  Just to clarify, in the included studies, was all treatment (including initial treatment) provided via telemedicine in the intervention groups of each study? From the inclusion criteria in the methods, this is my interpretation. If conventional care was provided in-person/face to face with telemedicine follow-up, were those studies excluded?  I am just wondering if the conclusion is that telemedicine a good alternative, or is it also a potentially cost-effective means to provide follow-up vs in-person counseling?

Author Response

Thank you very much for the valuable comments from the reviewers. The corresponding responses are as follows:

Point 1: Lines 51-52: By “lack of analysis of the overall outcomes” do you mean that not all potential outcomes have been evaluated in other meta-analyses? This needs a little more clarity.  

Response 1: "Overall outcome" refers to all potential outcomes, not just evaluations of indicators such as glycemic control and self-management abilities. As suggested, We will modify the content of line 52 and highlight it. Thank you for the suggestion.

Point 2: Line 55: I suggest changing “determined” to “evaluated”. Also, include the outcomes in the statement of purpose, e.g., “…. evaluated the effectiveness of telemedicine technology for ______ in the primary health setting.”

Response 2: We have revised as suggested as highlighted on lines 55-57. And add the result to the purpose section. Thank you for the suggestion. 

Point 3: Several columns in Table 1 could be combined to make a little more room: Author, Year, and Country could be in one column; Intervention time (suggest using duration) and frequency could be combined. The orientation of the table printout was portrait–perhaps the layout could be landscape to make more space for each column. It5’s awkward to read as it is now. The same applies to Table 2.

Response 3: As suggested, We have revised Table 1, put the author, year, and country together, and put together the intervention time and frequency of intervention, and adjusted the pages of Table 1 and Table 2 from vertical to horizontal for convenience read. Thank you for the suggestion.

Point 4: Please label all Figures as to what is being presented. One should be able to interpret the results presented in the figure without having to go to the text.

Response 4: As suggested, We added a description of the image content after each image and highlighted it. Thank you for the suggestion.

Point 5: Line 136: By “cross-theoretical model of change”, do you mean “trans-theoretical model of change”? If so, suggest changing.

Response 5: We modified the words in line 136 as suggested and highlighted them. Thank you for the suggestion. 

Point 6: Line 265 and Line 368: I suggest changing“confirms”to“supports”.

Response 6: This part has been revised as suggessted as highlighted on lines 270-397. Thank you for the suggestion.

Point 7: Lines 294-298: Did any other SRs and meta-analyses look at these outcomes? If so, were results consistent or different?

Response 7: We appreciate for your valuable comment. Based on your suggestion, We have added citations to articles similar to our findings at 306-307 and highlighted them. Thank you for the suggestion.

Point 8: Line 319: It would be helpful to include which outcomes differed in the previous studies. All of them or several of them?

Response 8: As suggested, We have added comparisons with other articles at 323-326 and highlighted. Thank you for the suggestion.

Point 9: Line 347:  What are the strengths of the study?  Suggest including these and changing heading to “Strengths and Limitations”.

Response 9: As suggested, We changed title 5 "Limitations" to "Strengths and Limitations" and added the strengths of this article in 388-393, highlighted. Thank you for the suggestion.

Point 10: Line 359: By differences in primary health care systems, do you mean what comprises conventional care, what the individual has to pay, other? Please clarify.

Response 10: The differences in primary health care expressed in this study mainly refer to: government funding for traditional health care. As suggested, we adjusted the content of lines 377-380 in the article, and indicated the main differences and highlighted them. Thank you for the suggestion.

Point 11: Line 361-362: It is unclear what is meant by“acceptance rate”. Of telemedicine? Is it participation or compliance to treatment? Both?

Response 11: "Acceptance rate" refers to the patient compliance rate for telemedicine intervention. As suggested, We have revised as suggested as highlighted on line 381.Thank you for the suggestion.

Point 12: Line 376: Just to clarify, in the included studies, was all treatment (including initial treatment) provided via telemedicine in the intervention groups of each study? From the inclusion criteria in the methods, this is my interpretation. If conventional care was provided in-person/face to face with telemedicine follow-up, were those studies excluded?  I am just wondering if the conclusion is that telemedicine a good alternative, or is it also a potentially cost-effective means to provide follow-up vs in-person counseling?

Response 12: In the included studies, the intervention group of each study provided initial treatment via telemedicine, the intervention was delivered via telemedicine, and there was no face-to-face contact during the intervention, but because laboratory testing of patients with type 2 diabetes required face-to-face blood draws for testing, So there was face-to-face contact during follow-up in some of the included studies. Therefore, based on the evidence we have, telemedicine interventions can only be proven to be a good complementary option. As suggested, we've made the changes as highlighted on line 403. Thank you for your question.

This manuscript is a resubmission of an earlier submission. The following is a list of the peer review reports and author responses from that submission.